Proteomic response of early juvenile Pacific oysters (Crassostrea gigas) to temperature

Crandall Grace 1
Elliott Thompson Rhonda 2
Eudeline Benoit 3
http://orcid.org/0000-0002-1791-2095 Vadopalas Brent 1
Timmins-Schiffman Emma 4
Roberts Steven 1 sr320@uw.edu
1 School of Aquatic and Fishery Sciences, University of Washington , Seattle, WA , United States
2 Mason County Public Health , Shelton, WA , United States
3 Taylor Shellfish Hatchery , Quilcene, WA , United States
4 Department of Genome Sciences, University of Washington , Seattle, WA , United States
Ward Eric
Electronic publication date: 2022 Oct 14
Publication date: 2022
Volume: 10
Electronic Location ID: e14158
Received 2021 Nov 30; Accepted 2022 Sep 8
Copyright: © 2022 Crandall et al.
Copyright year: 2022
Copyright holder: Crandall et al.
License: This is an open access article distributed under the terms of the Creative Commons Attribution License, which permits unrestricted use, distribution, reproduction and adaptation in any medium and for any purpose provided that it is properly attributed. For attribution, the original author(s), title, publication source (PeerJ) and either DOI or URL of the article must be cited.
License URL: https://creativecommons.org/licenses/by/4.0/

Keywords: Crassostrea gigas, Pacific oysters, Proteomics, Data-independent acquisition, Temperature, Ciliates

Funding: Washington Sea Grant Award NA140AR4170078 University of Washington Proteomics Resource UWPR95794 This work was supported by Washington Sea Grant award NA140AR4170078 and the University of Washington Proteomics Resource (UWPR95794). The funders had no role in study design, data collection and analysis, decision to publish, or preparation of the manuscript.

==============================
Pacific oysters (Crassostrea gigas) are a valuable aquaculture product that provides important ecosystem benefits. Among other threats, climate-driven changes in ocean temperature can impact oyster metabolism, survivorship, and immune function. We investigated how elevated temperature impacts larval oysters during settlement (19–33 days post-fertilization), using shotgun proteomics with data-independent acquisition to identify proteins present in the oysters after 2 weeks of exposure to 23 °C or 29 °C. Oysters maintained at elevated temperatures were larger and had a higher settlement rate, with 86% surviving to the end of the experiment; these oysters also had higher abundance trends of proteins related to metabolism and growth. Oysters held at 23 °C were smaller, had a decreased settlement rate, displayed 100% mortality, and had elevated abundance trends of proteins related to immune response. This novel use of proteomics was able to capture characteristic shifts in protein abundance that hint at important differences in the phenotypic response of Pacific oysters to temperature regimes. Additionally, this work has produced a robust proteomic product that will be the basis for future research on bivalve developmental processes.

Introduction

Oysters, such as the Pacific oyster (Crassostrea gigas), are a valuable aquaculture product and keystone species that provide essential ecosystem services within marine environments. Oysters are sessile organisms that form reefs that provide habitat for fish, invertebrates, and marine flora, and as filter feeders, oysters can have a positive influence on water quality (Coen et al., 2007; reviewed in Newell, 2004). Pacific oysters are broadcast spawners, releasing eggs and sperm into the water column where fertilization occurs. Larval settlement generally occurs 2 to 3 weeks after fertilization and requires the metamorphosis of free-swimming pelagic larvae to sessile juvenile oysters. Metamorphosis is an energy-intensive process during which larvae undergo complex behavioral and morphological changes.

As with many coastal marine organisms, changes in temperature can impact oyster physiology. In Ireland, mass mortality events of hatchery-produced adult Pacific oysters out-planted at two sites have been linked to high-temperature stress during the summer months with companion laboratory studies demonstrating decreased immune function at 21 °C compared to 12 °C by way of decreased phagocytosis in the hemolymph (Malham et al., 2009). Often, temperature changes are accompanied by changes in other environmental factors that can also have negative impacts. For example, Ko et al. (2014) found that low pH and low salinity combined with high temperature delays Pacific oyster larval growth rate before settlement and metamorphosis. These impacts of temperature can affect both cultured and wild oysters and are likely to be realized on a more frequent basis with climate-change-induced ocean warming (Helmuth & Hofmann, 2001).

While natural and uncontrollable temperature increases can be harmful, commercial hatcheries often rear oyster larvae in elevated temperatures under controlled conditions to achieve better outcomes. For many bivalves, increased temperature is often used to initiate spawning in the hatchery by mimicking spring conditions, when spawning naturally occurs (FAO, 2004). Increased temperature is also used to improve metabolism and growth in young oysters in hatchery settings. A comparison of larval physiology and early juvenile development of the Pacific oyster at five different temperatures (17 °C, 22 °C, 25 °C, 27 °C, and 32 °C) concluded that optimal growth rates and greatest settlement occurred at 27 °C (Rico-Villa, Pouvreau & Robert, 2009). Higher temperatures accelerate biological processes, including respiration and metabolism, allowing for faster development given that the larvae’s energetic needs are met, perhaps allowing young bivalves to “cruise” through the stressful metamorphosis period. However, there is likely a limit to any realized benefit with respect to temperature and duration of exposure.

Insight into molecular physiology at the protein level could help to explain the differences in larval success between different temperatures. The presence of certain proteins will show which genes are being translated and which proteins can be available for cell physiological function. An unbiased global proteomics survey can identify many of the proteins present in organisms at the time of sampling. This information can be leveraged to decipher the biological processes at play in various temperature treatments. Although it is still considered a novel approach, proteomics has been used in several studies of Pacific oyster response to a range of relevant environmental drivers and life stages (Venkataraman et al., 2019; Huan et al., 2012; Dineshram et al., 2012; Corporeau et al., 2012).

In this study, we used shotgun proteomics to understand how temperature impacts the physiology of recently settled oysters in a hatchery setting. We employed data-independent acquisition (DIA), which is a method of tandem mass spectrometry that achieves a more in-depth proteome coverage by acquiring all MS2 spectra for a given MS1 scan, unlike data dependent acquisition which only acquires the top n MS2 for each MS1 scan (Venable et al., 2004; Gillet et al., 2012). The goal of this experiment was to compare the proteomic responses of the oysters in the two temperature regimes, a typical rearing temperature of 23 °C and an elevated temperature thought to promote faster larval growth and better survival of 29 °C, and in the process develop a robust proteome that could build upon current Crassostrea gigas proteomic resources. These results contribute to our understanding of how temperature conditions impact oysters in hatchery settings, and suggest that oysters in the wild may be impacted by increasing temperature during the post-settlement stage.

Methods

Oyster rearing and treatment conditions

Adult oysters (n = 20) were strip spawned and eggs and sperm were combined for fertilization. Adult oysters (approximately 7 cm in size) were from Hood Canal, Washington. Oyster larvae were reared for 19 days post fertilization (dpf) before competent larvae were split between two silos, each a different temperature regime—conventional commercial conditions (23 °C) and elevated temperature (29 °C)—with 1.1 million Crassostrea gigas larvae reared in one fiberglass silo (46 cm diameter) per treatment. All seawater was pumped from Dabob Bay, WA, filtered through sequentially decreasing filter bags of 25, 10, and 5 µm, and treated with sodium carbonate (Na2CO3) to reach a pH set point of 8.4 (NBS scale). All oysters received the same mixed high-density microalgae diet (flagellates: Isochrysis spp., Pavlova spp., Nannochloropsis spp., Rhodomonas spp., and Tetraselmis spp.). Effluent algal densities were targeted at 100K cells ml−1. Crushed oyster shell (microcultch) graded from 180–315 µm was used as substrate for settlement and added to the oyster silos at 20 dpf. At 24 dpf (5 days after initiation of temperature treatment), oysters were screened to determine size and settlement rate, and then were returned to their respective silos. Settled oyster seed from each temperature regime was sampled for proteomic analysis on 24 and 27 dpf, after 5 and 8 days of temperature exposure. Dry oyster seed (approximately 500 µl) was flash frozen in liquid nitrogen prior to storage at −80 °C. The remaining seed was reared until 33 dpf, then screened to determine size and mortality (Fig. 1).

Figure 1 Experimental timeline.

At 19 days post-fertilization (dpf), Crassostrea gigas larvae were exposed to either 23 °C or 29 °C for 14 days. Proteomic samples were taken at 24 and 27 dpf, or days 5 and 8 of temperature exposure respectively. Settlement, growth, and mortality was assessed at 24 and 33 dpf, representing 5 and 14 days of temperature exposure, respectively.

Proteomic sample preparation

For proteomic analyses, four samples of pooled dry larvae (~250 ul equivalent) were prepared and analyzed representing two samples each from 23 °C and 29 °C, one from 24 dpf and one from 27 dpf. For each of the 4 samples, 50 mM ammonium bicarbonate (NH4HCO3) + 6M urea (500 µl) was added and larvae were homogenized using a pestle. Samples were centrifuged at 2,000g for 5 min. Supernatant (150 µl) was pipetted from each sample and placed into new tubes. The supernatant was sonicated three times each for 5 s at room temperature, cooling samples in between sonication rounds using an ethanol and dry ice bath for 5 s. Protein concentration was determined using a Pierce™ BCA Protein Assay Kit according to the manufacturer’s protocol, at 100 µg diluted in 50 mM NH4HCO3 (Thermo Fisher Scientific, Waltham, MA, USA).

Each sample (100 µg) was digested using trypsin and desalted for mass spectrometry as previously described (Timmins-Schiffman et al., 2017). Dried peptides were reconstituted in 100 µl 3% acetonitrile + 0.1% formic acid and stored at −80 °C. Data Independent Acquisition (DIA) was performed to assess protein abundance patterns via liquid chromatography tandem mass spectrometry (LC-MS/MS) (gradient of 2–35% acetonitrile over 60 min) with a Q-Exactive mass spectrometer (Thermo Fisher, Waltham, MA, USA). Full MS (MS1) settings included 60,000 resolution with 100 ms maximum injection time, and an AGC target of 3e6. DIA settings included 30,000 resolution, 1e6 AGC target, maximum injection time of 50 ms, loop count of 20, isolation window of 4 m/z, and an offset window of 0.0 m/z (Timmins-Schiffman et al., 2017).

Proteomic data analysis

Raw mass spectrometry files were converted to .mzML format using MSConvert from the ProteoWizard Toolkit version 3.0 (Chambers et al., 2012). Resulting files and the Crassostrea gigas deduced proteome (File S1) were used to create a chromatogram library using EncyclopeDIA with Walnut version 0.6.14 (Searle et al., 2018). Specific protocol details are provided in supplementary material (File S2). The chromatogram library, Crassostrea gigas proteome, and .mzML files were subsequently imported into Skyline Daily version 4.1.9.18271 (MacLean et al., 2010), which provides a means of setting filters, viewing spectral data for quality inspection, and exporting the data for downstream analyses (File S3).

Spectral data and proteins detected were exported for use in MS Stats (version 3.12.3, (Choi et al., 2014)). Within MS Stats, the two sampling dates (5 and 8 days of temperature treatment) were combined and treated as replicates to compare protein abundances between temperatures (File S4). Pooling the sampling dates provided a more robust analysis of the dominant trends in temperature response to compensate for the small number of samples. From the list of proteins, proteins that achieved the threshold of >2.00 and <−2.00 log-2 fold change (log2FC) were considered of interest for potential physiological importance, even though they were not significantly differentially abundant proteins (File S5). Specific protocol details are provided (File S6). DAVID, version 6.8 (Huang, Sherman & Lempicki, 2009a, 2009b), was used to identify the enriched Gene Ontology (GO) terms from the list of proteins of physiological interest in relation to all detected proteins. Additionally, enriched GO terms from the detected proteins were characterized in relation to the Crassostrea gigas proteome to capture the abundant biological processes present at this developmental stage, irrespective of temperature treatment, and enriched GO terms were identified using a <0.05 FDR cutoff.

Results

Phenotype

At 24 dpf (5 days into temperature treatment), oysters reared at 29 °C had a 22.6% settlement rate with a weighted average screen size of 560 µm. Approximately 25% of seed grown at 29 °C were 710 µm and larger. Oysters grown at 23 °C had a 9.2% settlement rate with a size of 363 µm at 24 dpf, with no seed exceeding 710 µm. At 29 dpf (10 days into temperature experiment), ciliates were visible at 23 °C. By 33 dpf, no oysters were alive at 23 °C, while survival of oysters grown at 29 °C was 86%.

Proteomics

There were 2,808 detected proteins (File S7), ~6.9% of the predicted proteins described in the published Crassostrea gigas proteome (File S8). Of the 2,808 detected proteins, 1,256 were associated with GO slim terms, the majority of which were related to metabolism, growth, and development (Fig. 2). These detected proteins were enriched for 108 biological process GO terms compared to the full, predicted Crassostrea gigas proteome.

Figure 2 Proportions of GOslim terms of all detected proteins.

Pie graph of the proportions of 1,256 detected proteins that fall within the Gene Ontology Slim terms listed in the legend: other biological processes; other metabolic processes; cell organization and biogenesis; developmental processes; transport; protein metabolism; RNA metabolism; signal transduction; stress response; cell cycle and proliferation; death; cell-cell signaling; cell adhesion; DNA metabolism.

Of the 2,808 detected proteins, 69 followed changing abundance trends that suggested physiological importance between the 23 °C and 29 °C treatments when using thresholds of >2.00 and <−2.00 log-2 fold change. Thirty-six proteins were higher in the 29 °C treatment, while 33 were higher in 23 °C treatment (Table 1). These proteins of interest contributed to 18 enriched GO biological processes (Fig. 3). Further analysis of this set of proteins identified enriched biological processes in the samples when compared with the full predicted proteome. Proteins higher in oysters grown in 29 °C were associated with biological processes primarily related to growth (Table 2), while those higher in oysters grown in 23 °C may contribute to the immune response (Table 3).

Table 1 Proteins of Physiological Interest.

Proteins identified using a <−2.00 log2FC and >2.00 log2FC cut-off. Proteins with log2FC values <−2.00 log2FC were more abundant in the oysters kept at 23 °C, while the proteins with log2FC values >2.00 log2FC were more abundant in the oysters kept at 29 °C.

Protein	Log2FC	E-value	Annotation	
CHOYP_CCDC146.1.1|m.11347	−6.8061724	0	Coiled-coil domain-containing protein 146	
CHOYP_AAEL_AAEL005639.1.1|m.10386	−4.8210246	0	ER degradation-enhancing alpha-mannosidase-like protein 2	
CHOYP_PCCB.2.2|m.62988	−4.4373158	0	Propionyl-CoA carboxylase beta chain, mitochondrial	
CHOYP_BRAFLDRAFT_120058.1.1|m.9895	−4.3588618	8.57E−20	DPY30 domain-containing protein 1	
CHOYP_LRRF2.2.2|m.40846	−4.0072733	3.63E−81	Leucine-rich repeat flightless-interacting protein 2	
CHOYP_BRAFLDRAFT_126379.1.3|m.5102	−3.7331439	1.09E−17	Tctex1 domain-containing protein 1-B (Fragment)	
CHOYP_BRAFLDRAFT_265162.1.3|m.58969	−3.4663681	3.11E−62	tRNA 2′-phosphotransferase 1	
CHOYP_LOC100377780.8.11|m.53259	−3.3450726	9.13E−41	Interferon-induced protein 44-like	
CHOYP_LOC100890099.1.1|m.11505	−3.2295286	2.99E−166	Actin, cytoplasmic 3	
CHOYP_LOC100748625.1.1|m.24479	−3.219579	1.35E−37	Nuclear pore complex protein Nup50	
CHOYP_LOC101174856.1.1|m.47585	−3.1898223	0	Mitogen-activated protein kinase kinase kinase 4	
CHOYP_LOC100372915.3.7|m.25794	−3.0750837	2.02E−102	Patched domain-containing protein 3	
CHOYP_ANR31.1.1|m.61075	−3.0736726	3.11E−13	Putative ankyrin repeat domain-containing protein 31	
CHOYP_ATS6.3.3|m.56109	−3.0371165	1.07E−43	A disintegrin and metalloproteinase with thrombospondin motifs 16	
CHOYP_LOC101160066.1.1|m.65775	−3.030505	3.18E−123	ADP-dependent glucokinase	
CHOYP_FASN.3.4|m.60278	−2.9470953	0	Fatty acid synthase	
CHOYP_ISCW_ISCW021350.1.1|m.10332	−2.9209024	7.71E−92	Protein SET	
CHOYP_IFIH1.10.14|m.60393	−2.9117919	1.89E−54	Interferon-induced helicase C domain-containing protein 1	
CHOYP_FUCO2.1.1|m.64139	−2.5948448	7.90E−178	Plasma alpha-L-fucosidase	
CHOYP_BRAFLDRAFT_81509.1.1|m.31565	−2.5764331	1.81E−42	Allograft inflammatory factor 1	
CHOYP_BRAFLDRAFT_87662.1.1|m.5474	−2.5756398	2.57E−151	N-acylglucosamine 2-epimerase	
CHOYP_ACTC.5.6|m.58242	−2.5403778	0	Actin, cytoplasmic	
CHOYP_TEKT5.1.1|m.60031	−2.5227287	2.55E−107	Tektin-3	
CHOYP_FCER2.8.9|m.27227	−2.4483624	3.69E−21	Perlucin	
CHOYP_DDX46.1.1|m.52473	−2.3581865	0	Probable ATP-dependent RNA helicase DDX46	
CHOYP_CRE_01395.2.2|m.44269	−2.3270263	2.35E−34	Tenascin-X	
CHOYP_NRDC.1.1|m.28526	−2.2979781	0	Nardilysin	
CHOYP_LOC410403.1.1|m.31535	−2.2622511	8.53E−95	Proteasome activator complex subunit 3	
CHOYP_LOC100709493.2.2|m.59425	−2.1893171	1.40E−68	Calpain-8	
CHOYP_BRAFLDRAFT_208293.6.20|m.36168	−2.1359859	3.04E−114	Heat shock 70 kDa protein 12B	
CHOYP_LYAM3.2.4|m.3333	−2.0863555	6.25E−18	P-selectin	
CHOYP_ISCW_ISCW014398.1.3|m.15838	−2.0670706	7.57E−84	Serine/arginine-rich splicing factor 1	
CHOYP_DNMT1.1.1|m.54990	−2.0263393	0	DNA (cytosine-5)-methyltransferase 1	
CHOYP_LOC100876012.1.1|m.8280	2.01458152	1.19E−95	Uridine phosphorylase 1	
CHOYP_PCKG.2.2|m.50139	2.04701648	0	Phosphoenolpyruvate carboxykinase [GTP]	
CHOYP_UNC89.1.19|m.7848	2.05672775	1.02E−38	Rab effector MyRIP	
CHOYP_LOC100377195.1.2|m.28624	2.06246173	1.95E−28	Mesenchyme-specific cell surface glycoprotein	
CHOYP_SCP.9.12|m.62606	2.0903096	2.29E−59	Sarcoplasmic calcium-binding protein	
CHOYP_BRAFLDRAFT_90098.2.4|m.5630	2.106258	5.32E−44	Betaine--homocysteine S-methyltransferase 1	
CHOYP_BRAFLDRAFT_227287.1.1|m.14669	2.14098269	5.42E−51	Metallo-beta-lactamase domain-containing protein 1	
CHOYP_TSP_03740.1.1|m.61194	2.14664447	1.24E−72	Peptidyl-prolyl cis-trans isomerase B	
CHOYP_PPA5.1.1|m.36087	2.15478979	2.67E−101	Tartrate-resistant acid phosphatase type 5	
CHOYP_LOC100024935.1.1|m.2615	2.1972962	6.34E−43	Thioredoxin, mitochondrial	
CHOYP_LOC100371209.1.1|m.49090	2.2051407	2.95E−108	LRP2-binding protein	
CHOYP_ARB_02478.1.1|m.29725	2.21514424	1.28E−39	Uncharacterized FAD-linked oxidoreductase ARB	
CHOYP_BRAFLDRAFT_76885.1.1|m.20259	2.24122259	6.79E−35	Gigasin-6	
CHOYP_NTF-2.1.1|m.38498	2.2417884	4.95E−30	Probable nuclear transport factor 2	
CHOYP_JNK.1.1|m.1540	2.24390832	0	Stress-activated protein kinase JNK	
CHOYP_BIRC6.1.2|m.51516	2.25147409	0	Baculoviral IAP repeat-containing protein 6	
CHOYP_BRAFLDRAFT_93126.2.3|m.15451	2.27030459	0	Mitochondrial Rho GTPase 1	
CHOYP_ZN706.1.2|m.11926	2.2779611	2.01E−29	Zinc finger protein 706	
CHOYP_TMEM2.4.4|m.28440	2.2865413	1.44E−99	Cell surface hyaluronidase	
CHOYP_LOC577383.1.1|m.8364	2.30433808	2.35E−65	RNA polymerase II-associated protein 3	
CHOYP_LOC100673983.1.1|m.10072	2.30829285	3.25E−128	Peroxisomal sarcosine oxidase	
CHOYP_VRK1.1.1|m.10917	2.35944728	2.61E−139	Serine/threonine-protein kinase VRK1	
CHOYP_LOC100374471.1.1|m.55948	2.42727851	1.89E−66	StAR-related lipid transfer protein 7, mitochondrial	
CHOYP_LOC100878123.1.1|m.55730	2.43195478	0	Procollagen galactosyltransferase 1	
CHOYP_BRAFLDRAFT_283451.1.1|m.41825	2.43609067	0	Integrator complex subunit 4	
CHOYP_BRAFLDRAFT_72357.1.1|m.59063	2.62270046	2.62E−101	Renalase	
CHOYP_DS.1.1|m.62106	2.64058171	0	Protein dachsous	
CHOYP_LOC100378619.2.2|m.36047	2.71093822	6.50E−40	Lamin tail domain-containing protein 1	
CHOYP_FAIM1.1.1|m.36651	2.74126935	8.46E−81	Fas apoptotic inhibitory molecule 1	
CHOYP_CLK2.1.1|m.3082	2.79615673	9.50E−156	Propionyl-CoA carboxylase alpha chain, mitochondrial	
CHOYP_NCBP2.1.1|m.60410	2.91829504	7.65E−86	Nuclear cap-binding protein subunit 2	
CHOYP_LOC100208382.1.1|m.12145	3.06964605	2.01E−22	Calmodulin	
CHOYP_GLOD5.1.1|m.25831	3.10869924	2.00E−61	Glyoxalase domain-containing protein 5	
CHOYP_CBR1.1.1|m.62756	3.45577173	3.73E−67	Carbonyl reductase [NADPH] 3	
CHOYP_contig_027593|m.31285	3.49679313	1.45E−146	Collagen alpha-2(I) chain	
CHOYP_HS90A.2.3|m.21992	3.57946239	0	Heat shock protein HSP 90-alpha 1	

Figure 3 Relationship between the enriched Gene Ontology biological process terms from the proteins of physiological interest found between larval oysters held at two different temperatures.

The size of the circle represents the fold enrichment, which demonstrates how enriched the process is in relation to the detected proteins. The color represents the different temperature treatments in which the gene ontology terms were more abundantly present. Lavender circles represent enriched processes that were higher in the 23 °C treatment, with larger circles being more enriched relative to the C. gigas proteome. Orange circles represent enriched processes that were higher in the 29 °C treatment, with larger circles being more enriched relative to the C. gigas proteome.

Table 2 Enriched GO terms of proteins that were higher in the 29 °C-exposed oyster seed than in the 23 °C seed.

Each row contains an enriched GO term ID, fold enrichment of the differentially abundant protein in relation to all detected proteins, and the function of the GO term.

Term	Fold enrichment	Function	
GO:0055123	14.37	Digestive system development	
GO:0035295	4.57	Tube development	
GO:0048568	7.66	Embryonic organ development	
GO:0007389	6.21	Pattern specification process	
GO:0045428	19.16	Regulation of nitric oxide biosynthetic process	
GO:0006809	19.16	Nitric oxide biosynthetic process	
GO:0009791	5.34	Post-embryonic development	

Table 3 Enriched GO terms of proteins that were higher in the 23 °C-exposed oyster seed than in the 29 °C-exposed seed.

Each row contains an enriched GO term ID, fold enrichment of the differentially abundant protein in relation to all detected proteins, and the function of the GO term.

Term	Fold enrichment	Function	
GO:0044712	6.68	Single-organism catabolic process	
GO:0044419	5.60	Interspecies interaction between organisms	
GO:0044403	5.60	Symbiosis, encompassing mutualism through parasitism	
GO:1901136	20.50	Carbohydrate derivative catabolic process	

Discussion

Using novel proteomics techniques, this study identified proteins of physiological interest when comparing larval Pacific oysters exposed to two different temperatures. While this study focuses primarily on hatchery applications, oysters can face temperatures of 23 °C and 29 °C in natural settings as well. As summarized in Lim et al. (2016), Pacific oysters can live in temperatures ranging 5–35 °C, and can handle acute exposure to 43 °C (Shamseldin et al., 1997; Lim et al., 2016). Temperature can have different effects on different stages of development; for example, the optimal range of temperature for gametogenesis is thought to be between 20–25 °C (Mann, Burreson & Baker, 1991; Lim et al., 2016). With tools like proteomics, we can get a deeper understanding of the mechanisms for different life stages, such as the development of oocytes (Corporeau et al., 2012), and with more research, investigate how different environmental conditions such as temperature impact the proteomic response of oysters. This has applications not only for developing optimal hatchery conditions for growth and production, but also for understanding how wild populations may fare in the face of climate change and other stressors.

Across the two temperature treatments, the detected proteins primarily represented those involved in metabolism, growth, and development. These findings are in agreement with the life history stage sampled, where growth rates are elevated and significant physiological changes are occurring related to somatic organization. In another study, researchers used in silico approaches to identify genes associated with larval settlement in Crassostrea gigas (Foulon et al., 2019). Approximately 27% of genes described by Foulon et al. (2019) had protein complements expressed in the current study. This not only validates the relevance of the developmental role of these proteins but also provides valuable resources for future work focused on metamorphosis and larval adhesion. Additionally, this comparison highlights the robust nature of the proteome developed as part of this study, along with the value of the Data Independent Acquisition proteomic approach.

Based on protein abundance comparison, we found that the GO Biological Processes detected in oysters in the 29 °C treatment were related to metabolism and growth, while the Biological Processes detected in oysters exposed to the 23 °C treatment were involved in immune system response. The presence of and interaction with ciliates in bivalve larval cultures have previously elicited a similar proteomic response in geoduck clam larvae (Timmins-Schiffman et al., 2020). Among the processes that were elevated at 23 °C in the oysters were Mitogen-activated protein kinase 4 (MAPK-4), which plays a role in the initiation of innate immune response (Krzyzowska et al., 2010), and allograft inflammatory factor 1 (AIF1), which has been shown to be a component in invertebrate immune response against pathogens and injury (Vizioli, Verri & Pagliara, 2020).

The growth and development of Pacific oyster larvae were positively impacted by exposure to 29 °C. The higher temperature likely promoted elevated metabolic rates, which in turn supported elevated growth and development. Higher temperatures (28–30 °C) have been shown to promote higher rates of metabolism and growth in another oyster species, Crassostrea corteziensis, in the juvenile spat life stage (Cáceres-Puig et al., 2007). Another possibility for the observed proteomic trend is that in the absence of other stressors, such as the ciliates observed at the lower temperature, there was an increased relative allocation of energy towards growth and development at the higher temperature. This is further supported by the phenotype data, which revealed that the oysters in the 29 °C treatment had higher settlement rates, greater size, no ciliates, and 86% survival, as compared to the 23 °C treatment group which had lower settlement rate, smaller size, and 100% mortality by the end of the experiment.

The oyster larvae samples from the 23 °C treatment had evidence of a greater physiological role of proteins associated with immune response when compared to the larvae at elevated temperature. At 29 dpf, ciliates were observed in the silo at 23 °C and by 33 dpf, all the oysters in the 23 °C treatment were dead. The predominant proteomic response was an immune response to parasites, supporting the idea that the oysters were initiating immune responses. Ciliate presence likely negatively impacted survival, either directly through parasitism or indirectly through increased energy allocation towards immune responses and away from critical maintenance processes. Ciliates have been a problem in hatcheries for decades and are associated with significant mortality events in early development bivalves (Elston et al., 1999). Ciliates may prefer colder temperatures or may not be able to survive at higher temperatures, protecting larval oysters at 29 °C against potential infections and the associated cost of launching an immune response. Alternatively, larvae may be physiologically compromised at lower temperatures, making them more susceptible to ciliates. In natural, non-hatchery settings, oyster susceptibility to ciliates increases with increasing salinity (Gauthier, Soniat & Rogers, 1990), and increases in summer and fall seasons when temperatures are roughly between 23 °C and 25 °C (McGurk, Ford & Bushek, 2016). This seasonal change observed could be related to laboratory findings where oysters were more susceptible to infection from OsHV-1 (oyster herpesvirus) in warmer temperatures, specifically 21 °C and 26 °C, though at an elevated temperature of 29 °C, the susceptibility of the oysters to OsHV-1 declined, and oysters at 29 °C had high survival rates (Delisle et al., 2018, 2020). Future research is needed to attempt to disentangle these phenomena and continue to elucidate factors contributing to improved survival in oysters.

Our findings can help hatchery workers, managers, and conservationists predict how temperature is and will impact oysters at the settlement stage in hatchery settings. The findings support an improved practice of increasing the temperature during the early developmental stage in the days preceding settlement to improve growth and survival. However further studies should investigate the optimal length of time and during which phase of development the larvae should be reared at elevated temperature. In addition, the annotated proteome developed as part of this work will be a valuable tool for future studies on bivalve development including providing specific targets for protein regulation studies in oysters as well as a reference for gene discovery in less studied bivalves.

Supplemental Information

Supplemental Information 1 Crassostrea gigas deduced proteome in .fasta format.

Click here for additional data file.

Supplemental Information 2 Protocol used for DIA analysis for this project- utilizing Walnut from the EncyclopeDIA suite.

Modified from the MacCoss Lab protocol. Creates chromatogram libraries from raw spectral data.

Click here for additional data file.

Supplemental Information 3 Skyline Daily protocol.

This protocol was used to view chromatogram DIA mass spectrometry results and to export results for analyses.

Click here for additional data file.

Supplemental Information 4 R script for using MS Stats to analyze the exported Skyline results, create figures, and to identify differentially abundant proteins.

For more information on how to use the script and MSStats package, see Skyline Daily protocol.

Click here for additional data file.

Supplemental Information 5 R script to identify significantly differentially abundant proteins.

MSStats did not find any significantly differentially abundant proteins using the default thresholds, so we used a threshold of <2.00 and >2.00 log2FC. This script was used to create a file that contains the differentially abundant proteins according to the <2.00 and >2.00 log2FC threshold.

Click here for additional data file.

Supplemental Information 6 Protocol for using MSStats to analyze exported Skyline Results.

Protocol that provides a bit more information on what was used for the R script for using MS Stats to analyze the exported Skyline results, create figures, and identify differentially abundant proteins.

Click here for additional data file.

Supplemental Information 7 Table of the 2,808 differentially abundant proteins.

The 2,808 differentially abundant proteins were identified in the R script to identify significantly differentially abundant proteins.

Click here for additional data file.

Supplemental Information 8 Table of 108 enriched biological process GO terms.

Table of the 108 enriched biological process GO terms, when comparing the list of 2,808 detected proteins (Table of the 2,808 differentially abundant proteins) to the full Crassostrea gigas proteome.

Click here for additional data file.

Additional Information and Declarations

Competing Interests

Author Contributions

Data Availability

The authors declare that they have no competing interests. Benoit Eudeline is employed by Taylor Shellfish Hatchery.

Grace Crandall analyzed the data, prepared figures and/or tables, authored or reviewed drafts of the article, and approved the final draft.

Rhonda Elliott Thompson conceived and designed the experiments, performed the experiments, prepared figures and/or tables, authored or reviewed drafts of the article, and approved the final draft.

Benoit Eudeline conceived and designed the experiments, performed the experiments, authored or reviewed drafts of the article, and approved the final draft.

Brent Vadopalas conceived and designed the experiments, authored or reviewed drafts of the article, and approved the final draft.

Emma Timmins-Schiffman conceived and designed the experiments, performed the experiments, analyzed the data, authored or reviewed drafts of the article, and approved the final draft.

Steven Roberts conceived and designed the experiments, analyzed the data, authored or reviewed drafts of the article, and approved the final draft.

The following information was supplied regarding data availability:

Additional files, scripts, and data are available at Zenodo: grace-ac, & Steven Roberts. (2021). grace-ac/paper-pacific.oyster-larvae: release for PeerJ submission (v2.1.0). Zenodo. https://doi.org/10.5281/zenodo.5708415.

The proteomic data is available at PRIDE: PXD015434.

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
