# Peer review of "Proteomic response of early juvenile Pacific oysters (Crassostrea gigas) to temperature"

_PeerJ, doi:10.7717/peerj.14158_

## Round 0.1 · original submission · Major Revisions

Both reviewers thought this manuscript is well written. In you revision, please pay attention to the comments from Reviewer 2 about the methods and statistical approach used.

·

Basic reporting

very good study ;please find my review in the attached document; several points must be detailed in material and methods, and discussion can be improved as mentionned in my comments.

Experimental design

well described and detailed enough.

Validity of the findings

good steps for increasing our knowledge on proteomic signals that concern larval development on bivalve molluscs.

Additional comments

I encourage such research studies that can help to emphasize new processes for improving health of larvae in mollusc hatcheries.

Reviewer 2 ·

Basic reporting

In this manuscript, the authors compares the proteomic response of the early juvenile Pacific oyster to two different temperatures (23°C and 29°C). The manuscript is well written and the methods they used (protein extraction, LC-MS/MS, label-freee quantification) are well suited for the study. However I find the analysis of the study, especially the statistical part, obscure and not really convincing. Furthermore the presentation of the results and the discussion is scarce and don't really give a full explanation.
In my opinion, it will be suitable for publication after a major change in the analysis, presentation of the study and an extended discussion. I also listed some minors changes below.

Experimental design

- On Oysters Rearing and Treatment Conditions :
If I understand the description right, there were only one silo per condition and no replicates ?

- On Proteomic sample preparation:
l.121 : I did not understand what "larval samples" stand for and why did you pool it ? Is there replicates of the protein samples ?
l.131-139: This method is not enough described. Which enzyme do you used for the digestion ? What was the gradient for the liquid chromatography ? What were the paramaters of the mass spectrometer ?

- On Proteomic data analysis :
l. 156-159: "significantly differentially abundant proteins" What was the test used ? What was the threshold, were the p-value corrected using FDR ?

Validity of the findings

A list of the proteins diferentially abundant should be given and not as supplemental data. The relatively short list of proteins (69 proteins) could then be discussed, if not one by one, at least more thouroughly thant just refering to GO biological process.

Additional comments

l.78-81 : this sentence is confusing and should be simplified
l.81-82: Identifying all proteins present in organisms are heavily dependent of the methods used, I would nuanced this sentence.
l.113 : precise the acronym "dpf" before using it

---

## Round 0.2 · Major Revisions

The reviewer notes a number of issues with the manuscript revision that was submitted. Like the reviewers, I'm a supporter of transparent / open science, but the paper as written does not actually reflect what was done, or the R scripts submitted. The reviewer recommends this paper be rejected; I am willing to consider one additional major revision -- but if the manuscript is not changed to reflect the modeling done, it will be rejected.

Reviewer 2 ·

Basic reporting

I had stated in my first review that “I find the analysis of the study, especially the statistical part, obscure and not really convincing. “I then asked for details in the specific comments
on how the list of "significantly differentially abundant proteins" were produced. (Asking :
What was the test used ? What was the threshold, were the p-value corrected using FDR ?) “

In the revised version of the manuscript, the paragraph was changed (l.165-170 of the current version 67862-v1) into the following :
“A list of detected proteins was determined using Tukey’s median polish test in MS Stats, and the p-values were corrected using false discovery rate (FDR). From the list of proteins, significantly differentially abundant proteins were identified using a threshold of >2.00 and <-2.00 log-2 fold change in RStudio (version 1.1.453, R Core Team, 2015) (Supplemental File 5). Specific protocol details are provided (Supplemental File 6).”

This paragraph is misleading on several levels. First the Tukey’s median polish is used during the processing of the MS raw data processing and not during the differential analysis as the sentence implied. Second, this is not a test and therefore it does not produce p-values. Third, as there were no p-values produced it is not possible that they were corrected using FDR. And finally, as there was no test used, there cannot be a list of “ significantly differentially abundant proteins”.
I dug a bit and took a look at the supplementary files. In SuppFile04, the R script used for the analysis, the groupComparison function from the MSstats was used for the differential analysis. This function uses a test based on a linear mixed effect model and produces adjusted-pvalues (using FDR) from this test. It’s later stated in the script that “The volcano plot for 23C-29C does not show the protein names because none of them is significant.”
Then the SuppFile05 is the script (as stated in the beginning) “ to subset the proteins that were detected with a log2FC greater than 2.00 and less than -2.00. These are differentially expressed, but not significantly.”
I have mixed feelings about the revised manuscript, on one hand I deeply appreciated that all the scripts for the entire analysis process were made available as it shows a completely transparent and great ethical way of doing science. On the other hand the methods description made in the manuscript is purposely misleading. In any cases, the list of proteins presented as “significantly differentially abundant proteins" are not significantly differentially abundant proteins. Moreover in my opinion it’s not possible to produce a list of “differentially expressed, but not significantly” proteins. If no statistical test is used but only a fold change threshold, the difference seen in the abundance of proteins can be due to chance. Therefore I do not recommend to publish this article, and I recommend the authors to find another way than using a differential analysis to publish this dataset.

Experimental design

see part 1

Validity of the findings

see part 1

Additional comments

see part 1

---

## Round 0.3 · accepted · Accept

I appreciate the work you've put into this revision. I think you have addressed all of the comments from the last review, and the new version is more clear. The reviewers and I appreciate making all the scripts for this paper available to follow what was done.